# Assessment of the Performance of Agricultural Tires Using a Mobile Test Bench

Roberto Fanigliulo, Marcello Biocca * , Renato Grilli, Laura Fornaciari, Pietro Gallo, Stefano Benigni, Paolo Mattei and Daniele Pochi

Consiglio per la Ricerca in Agricoltura e l'Analisi dell'Economia Agraria (CREA), Centro di Ricerca Ingegneria e Trasformazioni Agroalimentari, Via della Pascolare 16, Monterotondo, 00015 Rome, Italy
* Correspondence: marcello.biocca@crea.gov.it; Tel.: +39-06-90675215

**Abstract:** The performance of agricultural tires varies with the characteristics of both the terrain and the tractors on which they are mounted, which differently affect the rolling resistance, the traction capacity, and the slip. To reduce the variability of test conditions, CREA developed an original mobile test (MTB) bench which consists of a dynamometric single axle trailer pulled by a tractor and can be used both in traction performance tests (driving wheels) and in rolling resistance tests (driven wheels). A control system alternatively operates the adjustment of traction force or slip, so that each test is performed maintaining constant the desired values. The MTB underwent tests under different conditions (type of surface, pre-set values of force of traction and slip) aimed at verifying its accuracy and reliability. In a final test, two pairs of identical new tires were simultaneously mounted on the MTB and on the rear axle of the 2WD tractor that pulled it, to discover information on the different interactions occurring, under the same traction conditions, between the soil surface and each pair of tires, with reference to the relationship between the slips and the load transfers observed on the MTB and on the tractor rear axle. The results evidenced the capability of the MTB to guarantee repeatable test conditions, including field conditions, allowing comparison among the performance of different tires.

**Keywords:** tires tests; tractive performance; slip; dynamic load; rolling resistance; brake test; measurement sensors





## 1. Introduction

The study of the performance of agricultural tires particularly involves the properties of traction performance, slip, rolling resistance and the effects on soil compaction. Furthermore, aspects of safety and comfort have become important in recent years, considering the increasing speed limits allowed for tractors during transport on roads. The characteristics of agricultural tires and the instructions for their use and storage are annually published by the European Tire and Rim Technical Organization [1] and by ISO 11795:2018 [2]. Agricultural tractors are normally employed as power sources for pulling farm implements, through the conversion of engine power in useful drawbar power, thanks to the action of different typology tires with relative specific configurations (such as inflation pressure and wheel load), able to discharge the traction power on the ground. The tests on agricultural tires are commonly carried out at fixed points, using devices capable of adjusting the load conditions and simulating the contact with the tractive surface, or mounting the tires on vehicles they are designed for. In the first case, the tire testing conditions are often different from the real operational conditions; in the second case, the results of the tests are strongly affected by the interaction of factors not directly connected with the tires to test. The problem is further complicated by the variability of tire behaviour on various soil surface conditions. Consequently, the test conditions are rarely controllable and repeatable, often providing inconsistent results, even for the same tire model.

Regarding the test conditions and the adopted methodologies, international norms supply the procedures to follow in tire homologation activity aimed at the traffic circulation [3]. The aspects of tire rolling resistance and traction performances have been studied for many years and different approaches have been proposed [4–7]. In most of them, as said above, the tests are performed with the tires mounted on a vehicle pulling a second brake-vehicle generating a traction force that is measured together with the travel speed and the slip. The characteristics of the pulling vehicle affect the test results [8]. For instance, testing a model of tractor rear tire, the results are affected by several factors such as the construction characteristics of the tractor itself, the load transfer between the front and rear axle during the work, the front axle tire model, the presence of suspension systems, and the mechanical behaviour of a pneumatic tire [9]. The same tire model may perform differently when mounted on different tractors. Consequently, a reliable comparison among different tires having the same size would be possible only when mounting them all on the same tractor and the tire ranking observed could be validated by changing the tractor.

In order to reduce external influences during the tests on tires, different single wheel testers have been proposed to conduct tractive performance tests [10–13]. The University of Bologna developed a test bed consisting of a 70-m-long soil plot and of a hydraulically driven tool holder running on the plot at a maximum speed of 2.5 m·s$^{-1}$. Beyond the soil tillage tools, the tool holder can also support a wheel for studying the tire performance mainly from the point of view of soil compaction [14].

A similar "Terramechanics Rig" was developed at the Advanced Vehicle Dynamics Laboratory of Virginia Tech [15,16]. The test rig is a single-wheel tester that uses two separate motors to control wheel speed and slip of a tire that drives over a 7.62-m-long soil bin. A 6-axes wheel hub measurement system measures all moments and forces caused by the tire–soil interaction. Furthermore, the rig has normal load control to keep wheel load, applied with two pneumatic air springs that vary in air pressure and is controlled with an electro-pneumatic control valve that can hold several open and close positions proportional to an input voltage signal.

As to traction performances, a vehicle was produced in the USA and included a structure capable to support a wheel (or a track) and its ballast [17,18]. A hydraulic transmission drives the wheel in a soil bin and when its peripheral speed is greater than the travel speed, a positive slip and a thrust occur. A similar solution, pulled by a tractor, was proposed by Szente [19]. This is based on a single wheel vehicle supported by a frame that lodges the hydraulic system used to drive the wheel and adjusts the vertical load during the tests. Another solution driven in a similar way consisted of two vehicles that had to be alternatively utilized for wheels or tracks [20].

A more recent vehicle was utilized in the Firestone Test Centre in Columbiana, Ohio, USA. Its mass is 30,000 kg and it exerts over 151.240 kN of pull, simulating the resistance made by heavy farm equipment. It houses diagnostics equipment that enables the measurement of traction, drawbar pull, wheel slip, bar tear, and overall tire strength [21]. Czarnecki et al. [22] developed a test stand mounted on a two-part frame suspended on a three-point suspension linkage of a tractor. The tested wheel was mounted on a shaft and the wheel drive was obtained from the power take-off shaft through a reducing gear. The test stand covered studies on traction efficiency, slip, towing power and power lost on the rolling resistance and wheel slide.

Based on such experiences, CREA developed a simple and agile dynamometric, single-axle test bench, pulled by a tractor, for the tests of pairs of agricultural tires virtually under all conditions of soil surface and slope. This was called the Mobile Test Bench (MTB). This work describes the construction features of the MTB and the results of the tests carried out on it with the aim of contributing to the increase of the quality of measurement of the main parameters describing the performance of agricultural tires under operative conditions [23–25]. This should be possible by minimizing the effects of external factors and completely controlling the test conditions, so that they could become repeatable as well as the test results comparable. To reach this goal, the MTB provides the following

advantages: the collected data only refer to the tested tires; it accommodates different tire sizes and models; the total mass of the test bench is adjustable depending on the tire maximum load; the fundamental parameters (traction force, speed, slip, etc.) are adjustable and continuously monitored and recorded. It can be mainly employed in rolling resistance tests and traction tests of agricultural tires, under various surface conditions such as asphalt, concrete, field etc. The MTB underwent a series of efficiency tests aimed at comparing the interactions occurring between different soil surfaces and a pair of tires. Then, two pairs of identical tires were mounted both on the MTB and on the rear axle of a 2WD tractor which pulled the MTB [26], to investigate the relationship between the load transfers and the slips simultaneously and respectively observed on each pair of tires under the same traction conditions.

The aim of this paper was to describe and test the developed machine to ascertain whether the MTB is capable of providing accurate, reliable, and repeatable measurement of the main parameters describing the performance of agricultural tires. Consequently, it was used in tire testing activity, according to a suitable test protocol recognized by ENAMA [27], to assess the tractive performance under different soil conditions.

## 2. Materials and Methods

The mobile test bench (MTB) is comprised of a single-axle carriage for testing a pair of identical tires mounted with the rolling direction opposite to the travel direction. It is pulled by a traction vehicle (e.g., a tractor) and allows us to set the traction force values by means of a hydraulic braking system acting opposingly to the rotation of the wheels, which consequently slip. The main components of the MTB are described in the following.

### 2.1. Axle Gear Box Unit

The basic element of the MTB was the rear axle-gear box unit of a 127 kW power tractor. The gear box had 12 speed ratios and was equipped with a gear box main shaft, PTO-shaft, differential and differential lock. The rotational speeds of the wheels are not significantly affected by the presence of the differential. The other elements have been developed or assembled around this unit and their main characteristics are reported in Table 1.

**Table 1.** Main characteristics of the rear axle-gear box unit.

| Parameters | Values |
|---|---|
| Mass without oil, kg | 1200 |
| Mass with oil, kg | 1350 |
| Minimum width, mm | 2195 [1] |
| Maximum width, mm | 2575 [1] |
| Length, mm | 1800 |
| Mounted tires | 20.8 R 38 [2] |
| Minimum track, mm | 2245 [3] |
| Maximum track, mm | 2625 [3] |
| Power Take-Off, $min^{-1}$ | 540–1000 |

[1] Width at the end of the hub; [2] tires originally mounted on the tractor; [3] track with the originally mounted tires.

### 2.2. Hydraulic Braking System

The hydraulic braking system is a fundamental element in the tests on tire traction performance with the MTB (Table 2).

**Table 2.** Main characteristics of the hydraulic braking system.

| Parameters | Values | Type |
|---|---|---|
| Pump type | Gear | - |
| Pump displacement, L | 0.25 | - |
| Pump maximum speed, $min^{-1}$ | 4200 | - |
| Pump minimum speed, $min^{-1}$ | 800 | - |
| Pump diameter of the suction line, mm | 50.9 | - |
| Pump diameter of the delivery line, mm | 38.2 | - |
| Flow rate adjustment valve | - | Screw shaft |
| Flow rate control type | - | Electrical |
| Diameter of the shaft edge, mm | 38.1 | - |

Its function is to generate a controllable braking action on the wheels. Tire traction performance depends on the capacity to overcome this braking action, which is generated by a hydraulic circuit based on a 200 $dm^3$ oil tank, a suction line, a gear pump (connected to the gear box main shaft), a flow rate adjustment valve, and a delivery line. The oil used into the pump was a vegetable-based hydraulic fluid, to avoid possible oil leaks in the test environment. [28]. The braking unit (pump and flow rate adjustment valve) came from a dynamometric brake used in engine and PTO tests of tractors with power up to 130 kW. A 20 V DC, 40 A electric motor (maximum speed: 800 $min^{-1}$) operates the rotation of the adjustment valve shaft, whose stroke is defined by limit switches. The shaft position was detected by a potentiometric transducer having an interval of measurements of 0 to 10 V; its position can be determined manually, by means of a control knob or automatically by the feedback system basing on the signals provided by the sensors installed on the MTB. A motor reducer with a 14:1 transmission ratio was installed between the electric motor and the flow rate adjustment valve shaft.

The pump is driven by the wheel rotation through the gear box. The transmission ratio (Table 3) must be chosen on the basis of the test travel speed and must be compatible with the characteristics of the pump. When the flow rate adjustment valve is open, the oil is free to circulate. As it is progressively closed, the reduction of the delivery section and of the flow rate produce an oil pressure increase inside the pump, whose speed decreases. This decrease is transmitted to the wheels as a braking action. Consequently, the force needed to pull the MTB increases and the tires begin to slip.

**Table 3.** Transmission ratios between gear box and wheels.

| Gear Box Ratios | | Rotation Speed | | |
|---|---|---|---|---|
| No. | Name | Main Shaft, $Min^{-1}$ | Wheel, $Min^{-1}$ | Ratio, $\tau$ |
| 1 | 1 SR | 2122.0 | 6.3 | 334.95 |
| 2 | 2 SR | 2126.4 | 7.9 | 270.52 |
| 3 | 3 SR | 2124.2 | 9.6 | 220.81 |
| 4 | 1 R | 2140.6 | 13.1 | 163.28 |
| 5 | 2 R | 2123.3 | 16.1 | 131.87 |
| 6 | 3 R | 2124.6 | 19.7 | 107.63 |
| 7 | 1 N | 2120.8 | 24.6 | 86.08 |
| 8 | 2 N | 2124.7 | 30.6 | 69.52 |
| 9 | 3 N | 2125.4 | 37.5 | 56.75 |
| 10 | 1 V | 2124.2 | 50.6 | 41.96 |
| 11 | 2 V | 2124.1 | 62.7 | 33.89 |
| 12 | 3 V | 2124.8 | 76.8 | 27.66 |

*2.3. Cooling System*

The cooling system consists of an air-cooled heat exchanger (EMMEGI, type HPA52 Gr2) with an aluminium main body, receiving the delivery oil before it returns to the tank. The maximum oil flow rate is 260 L $min^{-1}$, as the maximum specific power dissipation is 1.7 kW. The airflow is provided by two fans driven by 12 V DC electric motors. The cooling

system is operated by a thermostat based on a thermocouple (type PT 100, class A, full scale: $-70 \div 200\,^{\circ}$C) measuring the oil temperature at the high-pressure pump outflow.

### 2.4. Ballast

The MTB has been conceived for testing different tires both for size and model. Among the characteristics of a tire, there is the maximum load specified for defined conditions of speed and inflation pressure. Consequently, the total mass of the MTB must be adjustable to reach the desired load value for the tire to be tested. The mass of the MTB is about 3500 kg. Additional mass is generally needed to reach the tire maximum load, according to the ETRTO specifications. The additional mass is represented by ballast elements having the shape and dimensions shown in Figure 1.

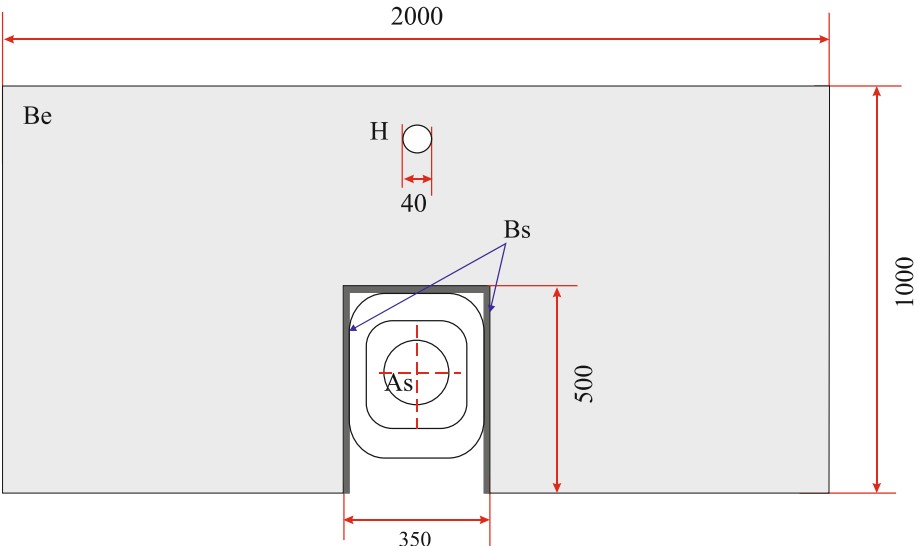

**Figure 1.** Schematic representation of the ballast. *Be*, ballast element; *Bs*, ballast support; *As*, axle-shaft; *H*, hooking hole. Unit: mm. *Be* thickness: 10 mm. *Be* mass: 138 kg.

Each element consists of a 10-mm-thick steel plate whose mass is 138 kg. Because of their symmetrical shape, the elements are positioned on the constant section supports realized on the right and left axle shafts, their centre of gravity lying in the same vertical plane of the wheel rotation axis. The available ballast mass is 8500 kg, allowing the MTB to reach a maximum mass of 12,000 kg, corresponding to a static load of 122.62 kN. Because of their dimensions, the supports can lodge a major mass. As said above, most of the mass (MTB and ballast) lies on the rotation axis: under full load static conditions (122.62 kN), the load measured on the tow-hook is 5.25 kN and the maximum vertical static load available for the tires is 117.37 kN. After its positioning, the ballast is fixed to the frame to avoid undesired shocks during the tests. The operative limits for tire testing with the MTB depend on the static load on its axle, which ranges from a minimum of 33.99 kN (MTB unballasted) up to 117.37 kN (MTB with maximum ballast). Since the MTB test concerns a pair of tires, these values must be divided by 2. Therefore, all tires whose load index corresponds to a load value in the range of 16.99 to 58.69 kN can theoretically be tested. In some cases, coupling between the rims and axle hubs will require appropriate adapters.

### 2.5. Measurement Sensors and Electronic Control System

The MTB is equipped with a series of sensors collecting the data of the test parameters and monitoring the working conditions of the hydraulic and cooling systems. The sensors for the basic parameters are the following:

- Timer, measuring the test duration;
- Digital encoder (250 pulses rev$^{-1}$) (Tekel TK510, Turin, Italy) on a reference free rotating wheel pulled by the MTB-tractor system, which measures the actual travel speed;

- Digital encoder (2500 pulses $rev^{-1}$) (Tekel TK510, Turin, Italy) measuring the peripheral speed of the tires mounted on the MTB wheels;
- 100.00 kN load cell (AEP Transducers TC4, Modena, Italy) on the coupling device, which measures the traction force (if necessary, it can be replaced with a 200.00 kN load cell);
- 50.00 kN load cell (AEP Transducers TC4, Modena, Italy) which measures the vertical load on the tow-hook.

As to the operating conditions, the main parameters are monitored by the following sensors:

- Digital encoder (100 pulses $rev^{-1}$) (Tekel TK510, Turin, Italy) measuring the pump speed;
- Pressure gauge, measuring the oil pressure into the pump (full scale: 30 MPa);
- Thermocouples, measuring the oil temperature into the tank and before the heat exchanger (type PT100, class A, full scale: $-71 \div 250\,°C$).

A data logger collects the signals from the sensors and sends them to a PC where they are processed. A feed-back system receives the signals corresponding to the measured values of the force of traction (or slip) and automatically adjusts the braking action to keep the parameter constant during the test according to the pre-set value.

From the basic measurements, the following parameters are calculated:

- Instant peripheral speed of the reference wheel, corresponding to the travel speed;
- Instant peripheral speed of the tested tire;
- Instant slip, calculated from the comparison of the two instant velocities;
- Pump speed.

All data were collected and processed in real time by an integrated system based on two units, a field unit and a support unit [29,30], fully assembled at CREA. The field unit consists of the tractor–MTB system and a photocell system (signalling the start and stop of the test) and is equipped with the cited transducers, a personal computer with a PCI card for real time data acquisition and an LCD monitor. A van equipped as a mobile laboratory represents the support unit. It is parked near the test site. Its PC communicates with the field unit's PC via a radio-modem system, exchanging data. The above-mentioned feedback system allows, both in the field unit and the support unit, to adjust in real time the traction force (or slip) and to monitor the trend of all measured parameters, including the safety ones. For each parameter, the acquisition system provided a frequency of acquisition of 10 Hz.

A second feedback system starts or stops the cooling system fans based on oil temperature. They are set to start working at 70 °C. The maximum admitted oil thermal leap is 10 °C. To avoid excessive thermal stress to the system, we decided to operate with oil pressure values lower than 10 MPa and pump speed values lower than 2000 $min^{-1}$ at the test velocities. When these values are reached, a safety system immediately opens the flow rate adjustment valve.

*2.6. MTB Frame*

The frame of the MTB (Figure 2) is made of H steel beams and is shaped as a cage around the axle-gear box unit. It comprises different parts designed to lodge the three different elements:

(1) The front section supports the pump-valve unit, the battery pack and the electronic control system and a parking wheel which is lifted during the tests. It terminates in a strong structure (whose height can be adjusted by means of a series of holes) supporting the towing eye. This is not connected to a common tow-hook which is instead inserted in a special coupling device sensorized by the two load cells mentioned in Section 2.5, protecting them from transversal shocks. The coupling device is adjustable in height and, despite being mounted in the rear part of the tractor, it is a conceptually integral part of the MTB. The described double height

regulation allows us to set the line of action of the traction force on the same horizontal plane passing for the centre of the wheels of the MTB.

(2) The central section is directly connected to the axle and has been dimensioned to support the mass of the oil in the tank and the ballast which rests on the wheel rotation axis.

(3) The rear section supports the cooling system.

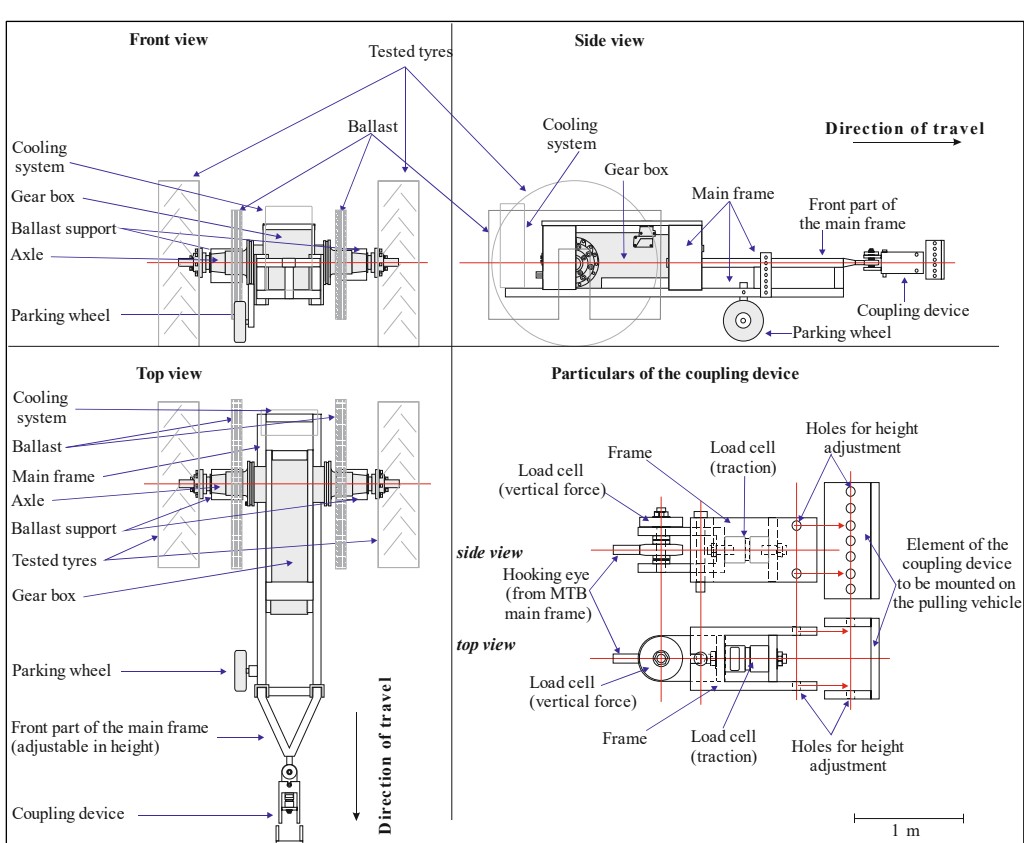

**Figure 2.** Schemes of the MTB frame and ballast and particulars of the coupling device.

*2.7. Traction Vehicles and MTB Functions*

Two tractors have been used to pull the MTB in the different series of tests: a 4WD 205 kW tractor (Case IH MX 270, Racine, WI, USA), with total mass of 11,000 kg (Figure 3a) and a 4WD, 107 kW tractor (Landini Legend 145, Fabbrico (RE), Italy), with total mass of 6420 kg and disengageable front wheel drive (Figure 3b).

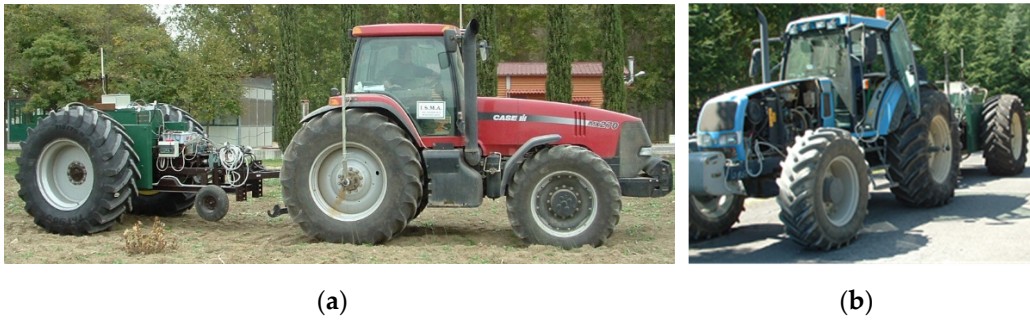

**Figure 3.** The MTB during the tests: (**a**) MTB on the compact surface with plant residue, pulled by a 205 kW tractor; (**b**) MTB and 4WD tractor with identical rear tires on asphalt track.

Before the tests, the engine performance of the two tractors was verified at the dynamometric brake (Borghi and Saveri, Bologna, Italy) which provided the updated characteristic curves to check the delivered power and the fuel consumption [31–33].

The MTB was mainly designed to investigate the characteristics of rolling resistance and traction performance of agricultural tires on different tractive surfaces [34]. The rolling resistance is directly provided by the values of the force of traction required to pull the MTB with the tested tires and their relative ballast [35]. The measurements performed by the load cell mounted on the drawbar are sufficient for this purpose. The hydraulic system is excluded by simply setting the gear box in the neutral position: in this case, the pump is disengaged, and the wheels can rotate without additional resistance.

The traction performance tests require the MTB braking action. After having defined the values of travel speed and traction force (or slip) to be applied, the more suitable gear box ratio must be chosen, to allow the pump to work within the fixed operating intervals of rotating speed (<2000 min$^{-1}$) and hydraulic pressure (<10 MPa). The intensity of the braking action is automatically determined by the feedback system. As for the normal tractor utilization in traction works, all the tests were performed on straight lines, with differential lock engaged, to obtain a more regular behaviour of the traction force. Unlike tractors, the traction on MTB is the consequence of a braking action and is exerted by the tires opposingly to the travel direction of the tractor–MTB system, which affects the aspects of slip measurement, tire mounting direction and dynamic load variations, which must be adequately investigated.

### 2.8. Theoretical Considerations

The meanings of the symbols used in this section are reported in the Legenda at the end of the text.

### 2.8.1. Travel Speed

The most severe traction conditions for agricultural tires depend on the characteristics of the ground surface and generally occur in the interval 0 to 2.8 m s$^{-1}$ (for example in primary soil tillage) and decrease for increasing velocities [36,37]. The tests were carried out within this interval. The travel speed was set by the tractor driver who pulled the MTB, based on the real time values provided by the reference wheel sensor.

### 2.8.2. Slip and Tire Mounting Direction

In consequence of the MTB braking action, the following considerations can be made. Considering a point P on the external surface of the tire, when the peripheral speed of the tire is equal to the travel speed, the relative speed between P and the ground surface during the contact is equal to zero. This means that there is no sliding of the soil along the lugs and the slip is s = 0.

In the case of the slip of a tractor's wheel, the tire peripheral speed, $v_{pT}$, is higher than the travel speed, v, and the difference $v_{pT} - v > 0$ represents the relative speed of the point P, $v_{RT}$, referred to ground surface, which means that during its contact with the tractive surface, P moves sliding rearward. In this case, the slip is calculated by means of the Equation (1):

$$s_T = 100 \frac{v_{pT} - v}{v_{pT}} = \frac{v_{RT}}{v_{pT}}, \tag{1}$$

The interval of variation of the $s_T$ is 0 to 100%. The maximum slip occurs when v = 0: the tractor is steady as the wheel keeps on rotating with peripheral speed $v_{pT}$. The vertex of the V-area of the lugs contacts the tractive surface before the end and the sliding of the soil along the lugs occurs from the vertex to the end of the V-area.

In the case of the MTB, due to the braking action, the travel speed, v, is higher than the tire peripheral speed, $v_{pMTB}$: the relative speed between P and the tractive surface is $v_{RMTB} = v - v_{pMTB}$. The direction of $v_{pMTB}$ is concordant with the travel direction, which

means that P is dragged forward and $v_{RMTB}$ is its speed with respect to the soil surface. In the present case, the tire slip on the MTB is calculated as follows:

$$s_{MTB} = \frac{v - v_{pMTB}}{v} = \frac{v_{RMTB}}{v} \tag{2}$$

Furthermore, $s_{MTB}$ ranges from 0 to 100%: when $v_{pMTB} = 0$ ($s_{MTB} = 100\%$), the MTB wheels stop rotating and are dragged on the tractive surface at travel speed v. When $s_{MTB} > 0$, if the tires are mounted with rolling direction opposite to the travel direction, the relative soil sliding along the lugs would occur from the vertex to the ends of the lugs, as in the case of tractor slip. Therefore, to restore on the MTB the interaction between tire surface and ground surface typical of tractors, the MTB's tires were mounted with rolling direction opposite to the travel direction.

The difference and relationship between $s_{MTB}$ and $s_T$ were investigated in a specific test in which the rear axle of the tractor Landini Legend 145 (see Section 2.5) and the MTB were equipped each with a pair of identical tires and the tractor was used as a 2WD tractor (front traction disengaged).

In such a system, during each test, the two pairs of tires underwent the same values of v and force of traction, $\overleftarrow{F}_{tra}$, as their peripheral speeds, $v_{pT}$ and $v_{pMTB}$, were measured. Under these conditions, the slips provided by the Equations (1) and (2) can be compared if the dynamic loads on the tractor rear axle, $\overleftarrow{Q}_{DT}$, and on the MTB, $\overleftarrow{Q}_{DMTB}$, are equal. In both relationships, the numerator is represented by the relative speed between the point P and the tractive surface: $v_{RT}$ for the tractor and $v_{RMBT}$ for the MTB.

If, under the same conditions of $\overleftarrow{Q}_D$, $\overleftarrow{F}_{tra}$ and v, we verified the equality:

$$v_{RT} = v_{RMBT} \tag{3}$$

this would indicate that the interaction between tire surface and ground surface is the same for the tractor tires and MTB's tires (mounted opposingly to the travel direction). Moreover, it would be $s_{MTB} > s_T$, because $v_{pT}$ is the denominator in (1), v is the denominator in (2) and it is $v < v_{pT}$. Eventually, it would be possible to fix a correspondence between $s_{MTB}$ and $s_T$, as follows:

$$s_T = \frac{s_{MTB} \cdot v}{v_{pT}} \tag{4}$$

In other words, using the MTB would provide a slip value ($s_{MTB}$) referring to the measured values of dynamic load, force of traction, travel speed and tire peripheral speed. If the Equation (3) is verified, from the $s_{MTB}$ values it would be possible to calculate the slip, $s_T$, which would occur if the same tires were mounted on a tractor and underwent identical conditions of dynamic load, force of traction and travel speed.

In the described test system, in the case of the MTB tires, the force of traction at the tow-hook, $\overleftarrow{F}_{tra}$, includes the rolling resistance, thus representing the gross force of traction. On the other hand, in the tractor, the force at the tow-hook is the net force of traction exerted by its rear tires which, in addition, also have to win the tractor motion resistance, $\overleftarrow{F}_{Tmr}$. This can be calculated as:

$$\overleftarrow{F}_{Tmr} = \overleftarrow{Q}_{DT} \ldots C_{mr} \tag{5}$$

where $\overleftarrow{Q}_{DT}$ is the dynamic load of the tractor during the traction tests; $c_{mr}$ is the motion resistance coefficient of the tractor which depends on the characteristics of the ground surface.

The gross force of traction exerted by the tractor rear axle tires, $\overleftarrow{F}_{traT}$, results from the relationship:

$$\overleftarrow{F}_{traT} = \overleftarrow{F}_{tra} + \overleftarrow{F}_{tmr} \tag{6}$$

In our case, $\overleftarrow{F}_{Tmr}$ was preliminarily determined on the test surface (asphalt), by directly pulling the tractor, with gear box disengaged, at the same speed value, v, adopted in the tests. From Equation (5), $c_{mr}$ was calculated from the ratio between the average force of traction (1.96 kN), and the tractor load (62.98 kN): = 0.0332. During traction tests, $c_{mr}$ was multiplied by the values of tractor's dynamic load, $\overleftarrow{Q}_{DT}$, to calculate the actual $\overleftarrow{F}_{Tmr}$. The sum of $\overleftarrow{F}_{tra}$ and $\overleftarrow{F}_{Tmr}$ provided the series of values of the gross force of traction exerted by the tractor rear tires. This series was compared to the $\overleftarrow{F}_{tra}$ series, referring to the MTB.

### 2.8.3. Dynamic Load

The vertical load to be applied on the tires depends on their characteristics. The maximum static load applicable in the MTB is 112.47 kN. When a force of traction is generated by the braking system, a proportional load transfer occurs, on the MTB, from the tires to the hooking eye. Therefore, the dynamic load on the MTB's tires decreases as the dynamic load on the tractor's rear axle increases. The load transfer on the sensorized coupling device is continuously measured by the load cell measuring the vertical force, $\overleftarrow{F}_v$, installed on it. The actual dynamic load on the MTB tires, $\overleftarrow{Q}_{DMTB}$, is given, instant by instant, by the difference between the MTB static load and the load transfer ($\overleftarrow{F}_v$). The ratio between force of traction, $\overleftarrow{F}_{tra}$, and $Q_{DMTB}$ provides the load utilization index, $I_{uQ}$:

$$I_{uQ} = \frac{\overleftarrow{F}_{tra}}{\overleftarrow{Q}_{DMTB}} \tag{7}$$

In a tractor, considering a given static load on the rear axle, the dynamic load, $\overleftarrow{Q}_{DT}$, considerably increases during the traction, for effect of the load transfer from the front axle, positively affecting the tractive performance [6,23]. Accurate measurements of the actual load transfer and dynamic load are very difficult on tractors and $\overleftarrow{F}_{tra}$ is commonly related to the static load, providing over-estimated values of $I_{uQ}$.

In a 2WD tractor, the load transfer from the front axle, $\overleftarrow{Q}_{tfa}$, can be approximately estimated by means of the Equation (7) [38,39]:

$$\overleftarrow{Q}_{tfa} = \frac{\overleftarrow{F}_{tra} \cdot h_{th}}{p} \tag{8}$$

where: p, pitch of the tractor (2750 mm in the tractor used in the tests); $h_{th}$, height of the tow-hook, decreasing as $Q_{DT}$ increases. As said before, in our tests the tow-hook is integrated in the coupling device and its height is 850 mm.

Considering the 2WD tractor–MTB test system, the following factors also must be considered to determine of the dynamic load on the tractor rear axle:

- A constant load, $\overleftarrow{Q}_{cd}$, determined on the tractor rear part, by applying the coupling device sensorized with the two load cells (0.54 kN). With respect to the rear axle, it is balanced by further a load transfer from the front axle, $\overleftarrow{Q}_1$, resulting from Equation (8):

$$\overleftarrow{Q}_1 = \frac{\overleftarrow{Q}_{cd} \cdot d_{TaRa}}{p} = 0.16 \text{ kN} \tag{9}$$

where: p, pitch of the tractor (2750 mm); $d_{TaRa}$, distance between tow-hook and rear axle (1000 mm). According to Equation (8), $\overleftarrow{Q}_1$ was equal to 0.16 kN and the resulting load increase on the rear axle caused by the MTB was $\overleftarrow{Q}_{cd} + \overleftarrow{Q}_1 = 0.704$ kN.

- The vertical force, $\overleftarrow{F}_V$, on the tow-hook, originated by the MTB load transfer during the tests. $\overleftarrow{F}_V$ varies during the test and is measured by the special load cell lodged in the coupling device. It is balanced by a load transfer, $\overleftarrow{Q}_2$, from the front axle of the tractor, that is calculated, instant by instant, by means of Equation (9):

$$\overleftarrow{Q}_2 = \frac{\overleftarrow{F}_V \cdot d_{TaRa}}{p} \tag{10}$$

where: p, pitch of the tractor (2750 mm); $d_{TaRa}$, distance between tow-hook and rear axle (1000 mm). In this case, the dynamic load increase on the tractor rear axle results as $\overleftarrow{F}_V + \overleftarrow{Q}_2$.

- The connection between the tow-hook and the main frame of the tractor is below the centre of rotation of the rear axle: the moment of the force of traction on it, is balanced by the moment of a load transfer, $Q_3$, from the front axle, as follows:

$$\frac{\overleftarrow{F}_{tra} \cdot d_{\Delta h}}{p} = Q_3 \cdot p \tag{11}$$

where: p, pitch of the tractor (2750 mm); $d_{\Delta h}$ (55 mm), difference between the height of the line of $\overleftarrow{F}_{tra}$ and the height of the rotation axis of the rear wheels, i.e., the under-load radius. In this case the load transfer is:

$$\overleftarrow{Q}_3 = \frac{\overleftarrow{F}_{tra} \cdot d_{\Delta h}}{p} \tag{12}$$

- The decrease of $Q_{DMTB}$ and the increase of $Q_{DT}$ during the traction determine a difference between the heights of the line of traction and of the centre of rotation of the rear wheels: the line of traction between the MTB and tractor is initially horizontal, then it slightly inclines forward due to the load transfers on both vehicles. The inclination has been measured to calculate the horizontal and vertical components of the force of traction. The vertical component, $\overleftarrow{F}_{traZ}$, is directed upward, and the corresponding decrease on the tractor rear axle results from:

$$\overleftarrow{Q}_4 = \frac{\overleftarrow{F}_{traZ} \cdot d_{TaRa}}{p} \tag{13}$$

where $d_{TaRa}$ is the horizontal distance between the tow-hook and rear axle. However, as $\overleftarrow{Q}_4$ is applied on the tow-hook, its measurement is comprised in the value of $\overleftarrow{Q}_2$ provided by Equation (10).

### 2.9. Preliminary Tests and Test Conditions

Preliminary tests were carried out with aim of drawing the curve of tire deflection under load conditions, used to assess the under-load tire radius during the tests with the MTB. In the test site (Figure 4a), the tires were mounted on a special axle whose main frame consists of a steel H-beam (overall dimension: 0.30 × 0.30 m, thickness: 30 mm). The centre of the axle was connected to a hydraulic cylinder lodged in a hollow in the ground and operated by a hydraulic pump. The hydraulic cylinder is capable of pulling the axle downward: the applied vertical force and the consequent tire deflection were respectively measured by means of a load cell placed on the traction rope (FN 1010-11-03, full scale: 200 kN) and a potentiometric transducer (OMAL PL400, full scale: 0–400 mm), which measures the axle lowering.

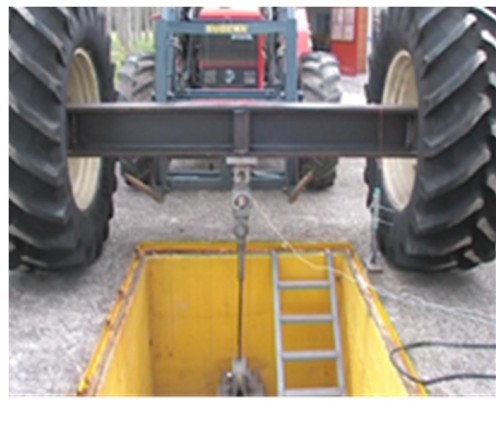
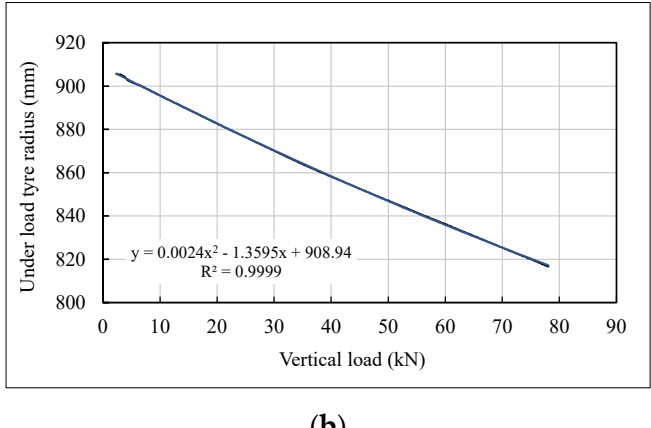

(**a**)　　　　　　　　　　　　　　　　　　　　　　　　　　(**b**)

**Figure 4.** Preliminary tests for the determination of the load-deflection curve: (**a**) Test plant: the tires are mounted on an axle realized on purpose; the load cell and the position transducer are visible. (**b**) Variations of the tire radius, as a function of the vertical load, obtained from the data of deflection, considering that at the inflation pressure of 0.16 MPa used in the test, the tire overall diameter is 1815 mm.

Considering that the deformation of the aforementioned H-beam (which constitutes the axle) stressed by the applied load is negligible, the change in tire deflection as a function of load provided the regression function visible in Figure 4b, which allowed us to assess the under-load tire radius from the values of the dynamic load, increasing the accuracy in the determination of tire peripheral speed and slip. Moreover, it was useful in the assessment of the component $\overleftarrow{Q}_3$ of the dynamic load on the tractor rear axle. Three series of tests (named A, B and C in the following) were carried out with the MTB, with tires at the inflation pressure of 0.16 MPa (Table 4).

**Table 4.** Test conditions.

| Parameters | Test A | Test B | Test C |
|---|---|---|---|
| Aim of the tests | MTB regulations; effects on different surfaces | Repeatability test | Study of slip depending on tire mounting direction |
| Test surface | Asphalt—seedbed—headland | Asphalt | Asphalt |
| Used tractor | 4WD 205 kW | 4WD 205 kW | 2WD, 107 kW |
| Gear box ratio [1] | 7–8–9 | 7–8 | 7–8–9 |
| Oil pressure values [2], MPa | 0–2.5–5.0–7.5–9.5 | - | - |
| Hydraulic pump speed [3], min$^{-1}$ | 2000 | 2000 | 2000 |
| Maximum temperature [4], °C | 60 | 60 | 60 |
| Test travel speed, m s$^{-1}$ | 1.94 | 1.39 | 1.53 |
| Tested radial tires | 520/70 R 38 | 650/75 R38 [7] | 650/75 R38 [7] |
| Inflation pressure [5], MPa | 0.16 | 0.16 | 0.16 |
| MTB static load [5], kN | 66.83 | 66.83 | 73.58 |
| Tractor rear axle static load [6], kN | - | - | 32.37 |

[1] Referring to Table 3; [2] Oil pressure values into the pump because of the reduction of the delivery. They have been repeated under different conditions (gear box ratios and soils) observing the consequent values of traction force and slip for estimating the MTB capability to perform the tests; [3] The maximum pump speed is 4200 min$^{-1}$; [4] The cooling system has been set to start when oil temperature reaches 50 °C; [5] The MTB static load and inflation pressure have been settled with reference to the standard requirements (ETRTO, 2013) for speeds up to 11.18 m·s$^{-1}$; [6] Foreseeing dynamic load increases up to match the MTB dynamic load; [7] Two couples of identical tires.

Test A aimed at verifying the capability and reliability of the hydraulic system to perform a suitable braking action and of the electronic system to maintain the oil flow rate and pressure inside the pump working interval and to monitor, record and process the data of the main test parameters. The test was carried out using a pair of new tires on three surfaces: asphalt track, headland with grass cover, and seedbed (surface prepared for the sowing). The latter was achieved through interventions of primary and secondary

soil tillage, according to normal farm practices [40,41]. As for the braking action exerted by the MTB, this resulted from the combination of hydraulic braking and the resistance to rotation exerted by the engagement of a given gear. The transmission ratio of each gear is therefore important: engaging low gears in the MTB results in high ratios and in greater resistance to rotation, which helps to reduce the hydraulic power required to achieve the desired tractive force. On the other hand, using low gears requires limiting the travel speed of the test system to avoid damage to the pump due to exceeding the operating speed limit. Various combinations of hydraulic oil pressure and gears were adopted in this series of tests, each time maintaining the selected oil pressure constant during the entire passage, in order to verify the behaviour of the MTB and investigate the effects of the same adjustment on the three test surfaces.

The second tests (Test B) were conducted on asphalt track with the same MTB–tractor system, to confirm the capability of the feedback system to automatically control the reference test parameters of force of traction and slip. The tests were carried on a reference distance (50 m) at a speed of 0.8 m s$^{-1}$, applying predefined values of force of traction, $\overleftarrow{F}_{tra}$, which had to be kept constant by the MTB feedback system during the entire passage. Test B also provided information of the variation dynamic load of the MTB, $\overleftarrow{Q}_{DMTB}$, and of the slip depending on $\overleftarrow{F}_{tra}$. The tests were replicated three times to obtain information about the repeatability of the results.

Test C refers to the considerations reported in Sections 2.8.2 and 2.8.3. This aimed at investigating the relationship between the slip on the MTB and on the tractor, the effects of the tire mounting direction and verifying the equality $v_{RT} = v_{RMBT_T}$ in connection with the behaviour of the dynamic loads. As in the Test B, it consisted of a series of passages (50 m) on asphalt track of the 2WD tractor–MTB system along a reference distance at a travel speed of 1.53 m s$^{-1}$ and at constant force of traction. In this case, the rear wheels of the tractor were sensorized with a digital encoder (2500 pulses rev$^{-1}$, Tekel TK510, Turin, Italy) which provided the data of their peripheral speed, allowing us to calculate their slip, $s_T$ according to Equation (4). At each passage, the setting value of the force of traction was increased in the range of 20 to 42 kN, by means of the MTB braking system. All the adjustments of the predefined speed and traction force values were made before the tractor–MTB system entered the reference base, in order to keep them constant along the entire distance. Data acquisition was started and stopped automatically by two photocells that delimited the reference base (see Section 2.4). As to the force of traction $\overleftarrow{F}_{tra}$ considered up to now, we referred to the traction performance of the MTB tires, while, for the tractor tires, we must consider the gross force of traction, $\overleftarrow{F}_{traT}$, resulting from the sum of $\overleftarrow{F}_{tra}$ and the tractor rolling resistance. As the tests were carried out in a plain, the $\overleftarrow{F}_{tra}$ variations due to the slope were not considered.

### 2.10. Data Processing

Travel speed, traction force, vertical force at the tow-hook and slip were the tire operative parameters involved in the tests. The data were processed in real time and their values monitored in the tractor cab, allowing the driver to adjust the travel speed and, in the case of the Test A, the braking action to keep the hydraulic oil pressure constant. The forces were directly measured by the load cells, while the different encoders provided the instant values of travel speed and peripheral speed of the tested tires, utilized for calculating the instant slip values, by means of Equation (2) for the MTB (Tests A, B, C) and (1) for the 2WD tractor (Test C). After each passage, the system provided the behaviours of said parameters along the reference distance. It also calculated their mean values, the standard deviations, and the coefficients of variation used to compare the results of the replications made with the same regulation and evaluating the test repeatability. Finally, aggregating the mean results from the different regulations allowed us to plot the variations of the forces (traction

and MTB load transfer) as functions of the slip and the variations of the load transfer on the tow-hook as a function of the force of traction.

Moreover, in Test C, the data on the vertical force on the tow-hook were used, in association with the load transfer values on the 2WD tractor, to assess the dynamic load and the gross traction force on the rear axle, respectively, by means of Equations (15) and (6). The data processing provided the diagram of the curves of dynamic load, for both tractor and MTB, as function of the gross force of traction. Similarly, a diagram was drawn for the curves of the relative speed, $v_{RT}$ and $v_{RMBT}$ (see Section 2.8.2) as a function of the gross force of traction. The intersections observed for the two pairs of curves represent the points in which, respectively, $\overleftarrow{Q}_{DT} = \overleftarrow{Q}_{DMTB}$ and $v_{RT} = v_{RMBT}$. If these points have the same abscissa value (force of traction), then Equation (3) would be verified.

Data analysis was carried out with the software MS Excel with descriptive statistics, Pearson's test of linear correlation, and linear regression.

## 3. Results and Discussion

### 3.1. Test A

Table 5 shows the values of slip and forces (average of three replications) obtained on the three test surfaces. Each passage of the test system was carried out while maintaining the constant oil pressure.

**Table 5.** Mean values of the test parameters (speed, slip and forces) observed during Test A with the MBT on the three test surfaces. (a) Asphalt; (b) seedbed; (c) headland with grass cover.

| Gear | Oil Pressure MPa | (a) | | | | | (b) | | | | | (c) | | | | |
|---|---|---|---|---|---|---|---|---|---|---|---|---|---|---|---|---|
| | | $v$ m·s$^{-1}$ | $s_{MTB}$ % | $s_{iMTB}$ % | $\overleftarrow{F}_{tra}$ kN | $\overleftarrow{F}v$ kN | $v$ m·s$^{-1}$ | $s_{MTB}$ % | $s_{iMTB}$ % | $\overleftarrow{F}_{tra}$ kN | $\overleftarrow{F}v$ kN | $v$ m·s$^{-1}$ | $s_{MTB}$ % | $s_{iMTB}$ % | $\overleftarrow{F}_{tra}$ kN | $\overleftarrow{F}v$ kN |
| - | 0 | 0.8 | 0.0 | 0.0 | 3.1 | 4.7 | 0.8 | 0.0 | 0.0 | 6.3 | 4.7 | 0.7 | 0.0 | 0.2 | 5.1 | 4.9 |
| 3N | 2.5 | 0.8 | 0.9 | 0.9 | 10.9 | 6.1 | 0.7 | 0.9 | 0.9 | 14.4 | 6.7 | 0.7 | 2.6 | 2.7 | 13.5 | 6.6 |
| 3N | 5.0 | 0.8 | 2.2 | 2.2 | 18.8 | 7.9 | 0.7 | 3.4 | 3.5 | 25.1 | 8.6 | 0.7 | 5.9 | 5.9 | 22.3 | 8.3 |
| 3N | 7.5 | 0.7 | 4.6 | 4.6 | 27.4 | 9.8 | 0.7 | 10.2 | 10.2 | 32.4 | 10.0 | 0.7 | 11.1 | 11.2 | 39.9 | 10.8 |
| 3N | 9.5 | 0.8 | 7.0 | 7.1 | 36.7 | 11.7 | 0.7 | 20.0 | 20.0 | 40.1 | 12.6 | 0.7 | 19.3 | 19.3 | 39.6 | 12.4 |
| 2N | 2.5 | 0.8 | 1.2 | 1.2 | 11.8 | 6.5 | - | - | - | - | - | - | - | - | - | - |
| 2N | 5.0 | 0.7 | 3.2 | 3.1 | 23.5 | 9.0 | 0.7 | 6.8 | 6.8 | 25.6 | 8.4 | 0.7 | 10.6 | 10.6 | 30.0 | 9.6 |
| 2N | 7.5 | 0.8 | 6.1 | 6.1 | 34.0 | 11.4 | 0.7 | 21.0 | 21.0 | 39.5 | 10.1 | 0.7 | 20.8 | 20.8 | 40.4 | 11.7 |
| 2N | 9.5 | 0.8 | 11.7 | 11.7 | 44.8 | 12.9 | 0.7 | 41.0 | 41.1 | 44.8 | 10.8 | 0.7 | 28.5 | 28.6 | 46.0 | 13.4 |
| 1N | 2.5 | 0.8 | 1.6 | 1.6 | 14.9 | 6.9 | - | - | - | - | - | - | - | - | - | - |
| 1N | 5.0 | 0.7 | 4.7 | 4.7 | 29.0 | 9.0 | - | - | - | - | - | - | - | - | - | - |
| 1N | 7.5 | 0.7 | 10.4 | 10.4 | 41.7 | 12.4 | - | - | - | - | - | 0.7 | 66.6 | 66.7 | 49.5 | 12.6 |
| 1N | 9.5 | 0.8 | 23.2 | 23.2 | 51.3 | 14.7 | - | - | - | - | - | - | - | - | - | - |

As for the slip, this is reported as "mean slip" (calculated from the mean travel speed and the mean peripheral speed of the MTB wheels along the test distance) and "mean instant slip", as the average of all the instant values of slip (10 values per second) calculated from the instant values of the travel speed and of the peripheral speed of the MTB wheels: their mean difference is 0.42%, indicating a good quality of instant slip measurements. It can be noticed that the intensity of the braking action is increased by increasing the oil pressure, but an important contribution is also given by the transmission: depending on the travel speed, choosing a lower gear box ratio causes higher mechanical resistance to the displacement of the MTB, allowing us to maintain lower oil pressure values. $\overleftarrow{F}_V$ represents the load transfer from the MTB tires to the tow-hook during the traction and has been used for calculating the actual dynamic load of MTB, $\overleftarrow{Q}_{DMTB}$, and the index of utilization of the load, $I_{uQ}$, as reported in Equation (6). The maximum mean values of the traction force (51.250 kN) and of the vertical force at the tow-hook (14.730 kN) have been observed on asphalt (gear: 1N; oil pressure: 9.5 MPa, Table 5): considering the MTB static load of 68.825 kN, the mean dynamic load on the tires during the test was 54.095 kN, resulting from the difference of the two vertical components. Again, according to Equation (7), the

ratio between the maximum traction force and the corresponding dynamic load is 0.947 and represents the index of utilization of the load, $I_{uQ}$, referring to the asphalt conditions.

Figure 5 reports some examples of the data series (instant slip and forces) provided by the system after each replication. They also show the different effects of the same regulation of gear box ratio and hydraulic oil pressure on the three surfaces: increasing the braking action, the slip increases more in conditions (b) and (c) than in (a). The curves show an increasing irregularity proceeding from (a) to (c), depending on the surface characteristics. In test (c), the grass cover heavily affects the soil–tire interaction determining sensitive instant slip variations. The shapes of the curves of the vertical force at the tow-hook suggest that this is related to the traction force, when the slip pattern is more regular as in (a), all conditions, in (b), 3 V-5.0 MPa and 2 V-7.5 MPa and in (c), 3 V-5.0 MPa.

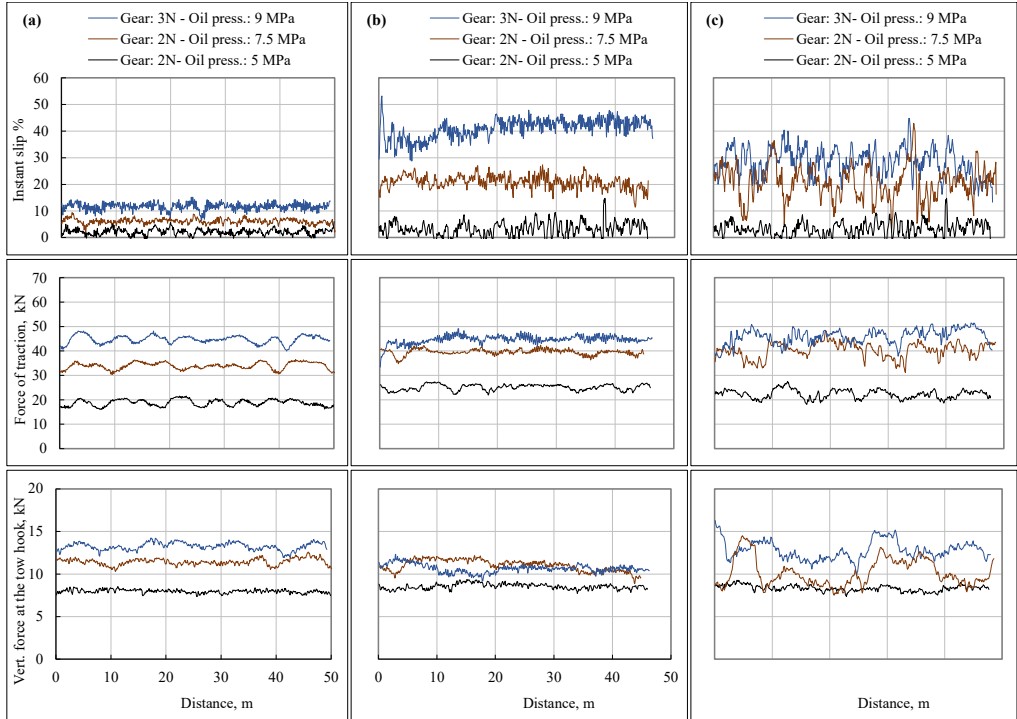

**Figure 5.** Effects of different regulations of the MTB braking system on the behaviours of slip, force of traction and vertical force on the tow-hook. (**a**) Asphalt; (**b**) seedbed; (**c**) headland with grass cover. Under constant conditions of gear box ratios and oil pressure, varying the test surface determines clear differences.

The results of the Pearson's test (Table 6) on the entire series of values of $\overleftarrow{F}_{Tra}$ and $\overleftarrow{F}_V$ of Figure 5 show higher correlations in the above-mentioned cases while, where the interaction between the ground and the tires is such as to determine large oscillations in the slip (uneven ground combined with increasing values of traction force), the correlation becomes gradually weaker and/or negative (although it persists with high *p* value), up to ceasing in (c), 2 V-9.5 MPa. Applying the Pearson test to the series of average values of $\overleftarrow{F}_{Tra}$ and $\overleftarrow{F}_V$ reported in Table 5, the variability caused by said oscillations is "bypassed" and the strong correlation between such parameters becomes evident in all test surfaces (Table 6).

**Table 6.** Results of the Pearson's tests carried out to assess the linear correlation between the force of traction, $\overleftarrow{F}_{tra}$, and the vertical force at the tow-hook, $\overleftarrow{F}_V$. Underlined p values indicate probability of correlation <0.001.

| Surface | Gear Pressure | r | *p* |
|---|---|---|---|
| (a) [1] | 3 V-5.0 MPa | 0.2921 | <0.001 |
| | 2 V-7.5 MPa | 0.7215 | <0.001 |
| | 2 V-9.5 MPa | 0.6538 | <0.001 |
| (b) [1] | 3 V-5.0 MPa | 0.3696 | <0.001 |
| | 2 V-7.5 MPa | 0.5162 | <0.001 |
| | 2 V-9.5 MPa | −0.3112 | <0.001 |
| (c) [1] | 3 V-5.0 MPa | 0.2225 | <0.001 |
| | 2 V-7.5 MPa | −0.3701 | <0.001 |
| | 2 V-9.5 MPa | −0.0115 | 0.783 |
| Correlation of means [2] | (a) | 0.9937 | <0.001 |
| | (b) | 0.9446 | <0.001 |
| | (c) | 0.9817 | <0.001 |

[1] Test executed on the entire series of values shown in Figure 5. [2] Test executed on all the mean values of $\overleftarrow{F}_{tra}$ and $\overleftarrow{F}_V$ reported in Table 5 for the three surfaces.

Figure 6 presents diagrams of the most significant parameters used in the evaluation of tire performance for each tractive surface. They have been drawn based on the average values of Table 5 ordered by increasing values of $\overleftarrow{F}_{tra}$, regardless of gear and oil pressure. They confirm the above observations from Figure 5 and Table 6. For a given slip value, the force of traction, $\overleftarrow{F}_{tra}$, increases passing from the most incoherent surface, (b) seed bed to the most compact and regular, (a) asphalt (Figure 6a). The MTB load transfer at the tow-hook, $\overleftarrow{Q}_{DMTB}$, has a similar behaviour. Furthermore, in Figure 6b, it can be noticed that the load transfer, $\overleftarrow{Q}_{tfa}$, varies linearly with the traction force with high coefficient of determination ($R^2$), following different behaviours on the three test surfaces. The ratio between the two forces probably represents, for each tire, a constant typical of each soil surface. Figure 6c shows that $\overleftarrow{Q}_{DMTB}$ as a function of slip, decreases as $\overleftarrow{Q}_{tfa}$ increases (Figure 6a) and this is more evident for increasing the compactness of the tractive surface. From the series of data of $\overleftarrow{F}_{tra}$, and $\overleftarrow{Q}_{DMTB}$, it has also been possible to draw the curve of the actual load utilization index, $I_{uQ}$ (Figure 6d).

*3.2. Test B*

Table 7 reports the results of the tests aiming to verify the reliability of the automatic control system and its capability to replicate the tests. The force of traction was the reference parameter: it has been set as the values of 32 and 36 kN and the feedback system operated for keeping them constant.

It can be noticed that the mean values of the force of traction obtained are very similar to the setting values and that the differences among the replications are very small, depending on the variability of the characteristics of the asphalt track sections where the replications were carried out, as testified by the low values of the coefficients of variation (CV) of the averages.

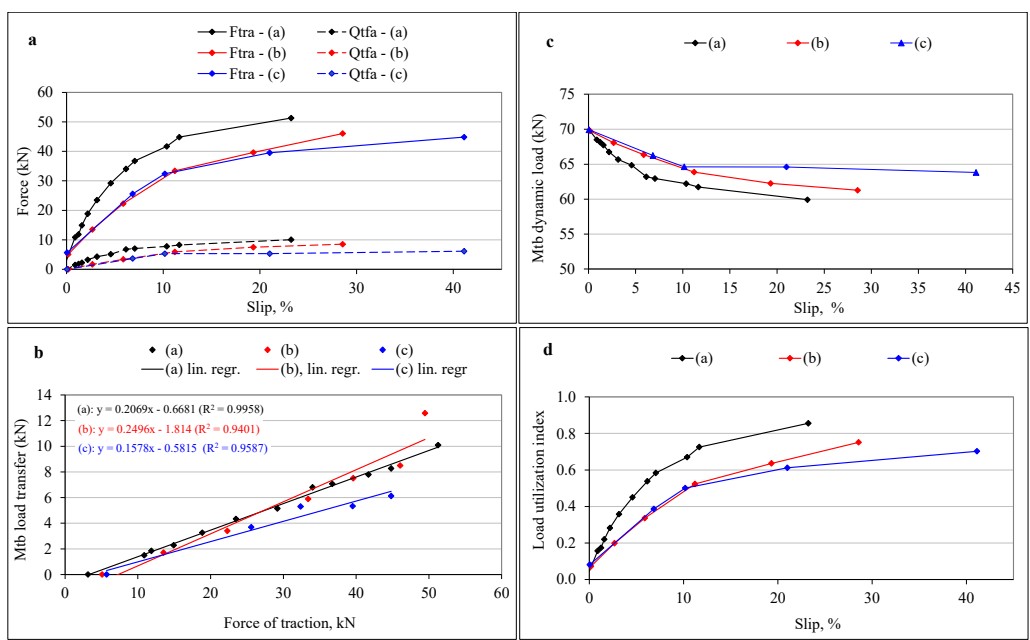

**Figure 6.** Results of test A on the three surfaces; (**a**) force of traction, $\overleftarrow{F}_{tra}$, and load transfer, $\overleftarrow{Q}_{tfa}$, on the tow-hook as functions of the slip; (**b**) linear regression curves of the variation of $\overleftarrow{Q}_{tfa}$ as a function of $\overleftarrow{F}_{tra}$; (**c**) decrease of MTB dynamic load, $\overleftarrow{Q}_{DMTB}$, during the tests (difference between static load and load transfers); (**d**) variation of the index of load utilization, $I_{uQ}$, as a function of the slip.

**Table 7.** Statistical indicators of the main parameters measured by the MTB for the pre-set values of force of traction of 32 and 36 kN.

| Parameters | Statistical Indicators | Pre-Set $\overleftarrow{F}_{tra}$: 32 kN | | | | | | Pre-Set $\overleftarrow{F}_{tra}$: 36 kN | | | | | |
|---|---|---|---|---|---|---|---|---|---|---|---|---|---|
| | | Replications | | | Average | St. dev. | C.V. | Replications | | | Average | St. dev. | C.V. |
| | | 1 | 2 | 3 | | | | 1 | 2 | 3 | | | |
| Traction force | Average, kN | 31.91 | 31.91 | 32.10 | 31.98 | 0.10 | 0.32 | 35.96 | 35.99 | 36.01 | 35.98 | 0.02 | 0.06 |
| | St. dev., kN | 0.74 | 0.66 | 0.89 | 0.76 | - | - | 0.73 | 0.58 | 0.79 | 0.70 | - | - |
| | C.V. | 2.31 | 2.05 | 2.77 | 2.38 | - | - | 2.03 | 1.62 | 2.19 | 1.94 | - | - |
| Load transfer | Average, kN | 13.38 | 13.90 | 11.73 | 13.00 | 1.13 | 8.69 | 14.04 | 14.05 | 14.37 | 14.15 | 0.19 | 1.35 |
| | St. dev., kN | 0.37 | 0.28 | 0.33 | 0.33 | - | - | 0.26 | 0.26 | 0.28 | 0.27 | - | - |
| | C.V. | 2.78 | 2.04 | 2.84 | 2.55 | - | - | 1.85 | 1.86 | 1.95 | 1.89 | - | - |
| Mean slip, (%) | | 6.11 | 5.48 | 5.48 | 5.69 | 0.36 | 6.29 | 7.30 | 7.40 | 7.75 | 7.48 | 0.24 | 3.17 |
| Instant slip | Average, % | 6.10 | 5.51 | 5.57 | 5.73 | 0.33 | 5.71 | 7.29 | 7.40 | 7.73 | 7.48 | 0.24 | 3.16 |
| | St. dev, % | 0.64 | 0.61 | 0.54 | 0.59 | - | - | 0.60 | 0.65 | 0.69 | 0.69 | - | - |
| | C.V. | 10.47 | 10.99 | 9.67 | 10.38 | - | - | 8.19 | 8.84 | 8.85 | 8.63 | - | - |

Moreover, similar load transfer and slip values occur in the replications with the same setting of force of traction. Eventually, the averages of the instant slip, $s_{iMTB}$, and of the mean slips were always very similar, although the instant slip showed higher standard deviations and CVs due to the instant oscillations discussed in Test A (at pressure priority). However, in Test B, $s_{iMTB}$ and $s_{MTB}$ showed lower amplitude and variability because of the strict control exerted by the feedback system on the force of traction. In confirmation of this, for instance, in the condition "2N-95 MPa" of Test A (Figure 5a), we observed a mean $\overleftarrow{F}_{tra}$ of 34.00 kN, an intermediate value between 31.98 and 35.98 kN reported in Table 7, with $s_{iMTB}$ = 6.13% and $s_{MTB}$ = 6.6%. The standard deviation and the CV of $s_{iMTB}$ were, respectively, 0.99 and 16.2%, higher than the values shown in Table 7 for both conditions there considered. Similar results were obtained in tests in which the feedback signal was represented by the slip values, testifying the good accuracy and reliability of the test system.

*3.3. Test C*

The results of Test C are reported in Figure 7. As in Tests A and B, the dynamic load of the MTB, $\overleftarrow{Q}_{DMTB}$, resulted directly from the difference between the static load and the load transfer, while the dynamic load on the tractor rear axle, $\overleftarrow{Q}_{DT}$, was assessed by means of Equation (15) after having determined its components according to Equations (7) to (14). Regarding the gross force of traction, $\overleftarrow{F}_{tra}$ was considered for MTB tires, as $\overleftarrow{F}_{traT}$ was calculated for rear tractor tires according to Equation (6). In our case, the tractor resistance to motion, $\overleftarrow{F}_{Tmr}$, was preliminarily determined on the test surface (asphalt), by directly pulling the tractor, with gear box disengaged, at the same speed value, v, adopted in the tests. From Equation (5), $c_{mr}$ was calculated from the ratio between the average force of traction (1.96 kN), and the tractor load (62.98 kN): $c_{mr}$ = 0.0332. During traction tests, $c_{mr}$ was multiplied by the values of the tractor's dynamic load, $\overleftarrow{Q}_{DT}$, to calculate the actual $\overleftarrow{F}_{Tmr}$. The sum of $\overleftarrow{F}_{tra}$ and $\overleftarrow{F}_{Tmr}$ provided the series of values of the gross force of traction $\overleftarrow{F}_{traT}$, exerted by the tractor tires, which, in Figure 7, represented the abscissas of $\overleftarrow{Q}_{DT}$ and $v_{RT}$, while the series of $\overleftarrow{F}_{tra}$ was used as the abscissa for $\overleftarrow{Q}_{DMTB}$, and $v_{RMTB}$.

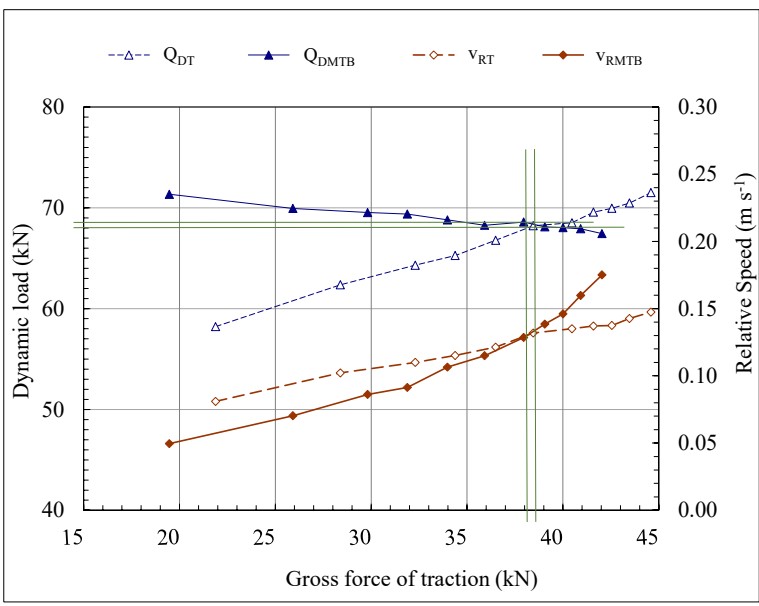

**Figure 7.** Trend of the tire dynamic loads, $Q_{DT}$ and $Q_{DMTB}$, and of the relative speed, $v_{RT}$ and $v_{RMBT}$, between tire surface and test track surface, respectively, observed for the MTB and the rear tires of the 2WD tractor.

In the intersection points between the two pairs of curves, where $Q_{DT}$ = $Q_{DMTB}$ and $v_{RT}$ = $v_{RMBT}$, we can observe a difference of about 0.7 kN between their abscissas (1.3% of the gross force of traction for $Q_{DT}$ = $Q_{DMTB}$). A similar difference between the values of $Q_{DMTB}$ and $Q_{DT}$ can be observed in correspondence with the gross force of traction value for $v_{RT}$ = $v_{RMBT}$ (about 0.5 kN, 0.08% of the tractor dynamic load). These differences, however small, can be explained by a residual indeterminacy in the assessment of the components of the tractor dynamic load. In this context, Equation (3) can be considered verified and, consequently, the relative speed ($v_{RT}$ = $v_{RMBT}$ = $\Delta v$) between the tire surface and the track surface is equal for the MTB and the tractor under the same conditions of dynamic load, travel speed and gross force of traction.

Figure 8 shows the different trends of slip for tractor and MTB, respectively calculated by means of Equations (1) and (2) relating to the test travel speed: $s_{MTB}$ and $s_T$ clearly differ for the same $\Delta v$ values. For a $\Delta v$ = 0.4 m·s$^{-1}$, we can observe that $s_{MTB}$ = 28.8%, as $s_T$ = 22.36%.

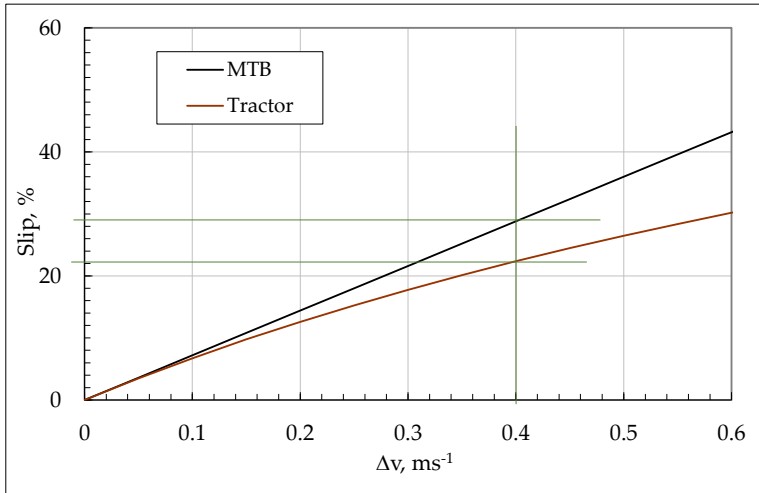

**Figure 8.** Slip values in the MTB tires, $s_{MTB}$, and in the 2WD tractor rear tires, $s_T$, as a function of the relative speed between tire and soil. The diagram refers to the test speed of 1.39 m·s$^{-1}$, but a similar behaviour can be observed for each speed.

Based on the values of gross force of traction, slip, dynamic load and travel speed provided by the MTB for a couple of tires, by means of Equation (4), it is possible to calculate the slip they would have if they were mounted on a tractor exerting the same gross force of traction at the same travel speed, with the same dynamic load. According to these considerations, the only consequence caused by the difference between the two conditions (i.e., MTB tire peripheral speed lower than tractor tire peripheral speed) is that, along a given distance, the number, N, of contacts between a point P on the tire surface and the soil is lower for the MTB than for the tractor, while the duration, t, of each contact is longer. In the limit case of s = 100%, in MTB, P remains stationary, and the tire is dragged on the ground (it is as if the ground moves backward under the same point P of the tyre), while in the tractor the tire continues to rotate (P moves) on the same point of the ground that is perfectly still. This case is, however, far from normal working conditions, where the slip ranges between 0 and 40%. Within this interval, traveling a given distance at a given speed, the total time, T, of contact between P and the soil, results as T = Nt and T can be considered the same for the MTB and the tractor. The different number and duration of contacts could have some different thermal effects relating to tire wear, but it does not seem probable that, under normal conditions of soil surface characteristics, temperature, and humidity, said differences could become significant. The data processing system of the MTB can be settled for directly showing the presumed tractor slip values based on travel speed and the dimensional characteristic of the tires under test.

## 4. Conclusions

The results of the tests on the Mobile Test Bench (MTB), developed for agricultural tire testing under real working conditions, demonstrated the capability of the prototype to carry out tests on tire tractive performance. Compatible with its design features, the MTB allows testing of a wide range of agricultural tires under all conditions of surface and slope, overcoming the limitations of the majority of the already proposed solutions. Moreover, it allowed us to solve the problems related to control and direct detection of dynamic vertical load on the tires being tested, which remained largely undetermined when the tests were carried out with tires mounted on tractors. Both braking systems and electronic control systems showed high accuracy in measurements and data processing, as the feedback system was effective and precise in continuously controlling the main test parameters (force of traction or slip). The consequent high repeatability of the test conditions allows accurate comparison of the performances of different tire models and reliable assessment of tire design effects on energy dissipation and savings.

As for the differences in tire–soil interaction and dynamic load behaviour between the tires mounted on the MTB and the tractor, the test results confirmed the hypothesis that, under normal conditions of use, the relative movements and speeds between tire and track surface are similar in both cases. Therefore, from the information provided by the MTB on a tire model, it is possible to derive the performances referring to the same tire mounted on a tractor.

The Mobile Test Bench can be used both in tire rolling resistance tests and in braking tests for tractor traction tests (according to OECD Code 2), taking care that, in these cases, the direction of mounting of the tire coincides with the direction of travel. The MTB could also contribute to investigating aspects of noise emissions and vibration levels determined by tires and their effects on soil compaction, reducing the disturbance of external factors (e.g., noise and vibrations emitted by the engine) on the measurements.

**Author Contributions:** Conceptualization, D.P.; methodology, D.P. and R.F.; validation, D.P., M.B. and R.F.; formal analysis, P.G., R.G., S.B. and L.F.; investigation, D.P. and R.F.; resources, R.F.; data curation, D.P., R.G., S.B., P.M. and R.F.; writing—original draft preparation, D.P. and R.F.; writing—review and editing, R.F., L.F. and M.B.; visualization, M.B. and P.G.; supervision, D.P.; project administration, P.G. and P.M.; funding acquisition, D.P. and R.F. All authors have read and agreed to the published version of the manuscript.

**Funding:** This research was funded by the Italian Ministry of Agriculture (Mi.P.A.A.F.) under the AGROENER project (D.D. n. 26329, 1 April 2016)—http//agroener.crea.gov.it/ (accessed on 10 December 2021).

**Institutional Review Board Statement:** Not applicable.

**Data Availability Statement:** Not applicable.

**Acknowledgments:** Special appreciation to Gino Brannetti and Cesare Cervellini (CREA) for their collaboration in the realization and assembling of the elements of the prototype and in the field tests.

**Conflicts of Interest:** The authors declare no conflict of interest.

## Abbreviations

| | |
|---|---|
| $v$ | travel speed measured by means of a reference idle wheel trailed by the test system (m s$^{-1}$). |
| $v_{pT}$, $v_{pMTB}$ | peripheral speed of tractor (rear axle) and MTB wheels with slip (under-load radius) (m s$^{-1}$). |
| $v_{RT}$ | relative speed of a generic point P on the surface of tractor rear tire, referred to the ground (m s$^{-1}$). |
| $v_{RMTB}$ | relative speed of a generic point P on the surface of MTB tire, referred to the ground (m s$^{-1}$). |
| $s_T$ | tractor slip (%). |
| $s_{MTB}$ | MTB slip (%). |
| $s_{iMTB}$ | MTB instant slip (%). |
| $\overleftarrow{F}_{tra}$ | MTB traction force at tow-hook (kN). |
| $\overleftarrow{F}_{traT}$ | gross traction force exerted by the tractor (kN). |
| $\overleftarrow{F}_{traZ}$ | vertical component of the $\overleftarrow{F}_{tra}$ (when $F_{tra}$ is not horizontal) (kN). |
| $\overleftarrow{F}_V$ | vertical force measured at the tow-hook between tractor and MTB (kN). |
| $\overleftarrow{F}_{mr}$ | tractor motion resistance (kN). |
| $\overleftarrow{Q}_{ST}$ | static load on tractor's rear axle (kN). |
| $\overleftarrow{Q}_{DT}$ | dynamic load on tractor's rear axle under traction conditions (kN). |
| $\overleftarrow{Q}_{SMTB}$ | static load on MTB axle (kN). |
| $\overleftarrow{Q}_{DMTB}$ | dynamic load on MTB axle under traction conditions (kN). |

| | |
|---|---|
| $c_{mr}$ | coefficient of motion resistance. |
| $I_{uQ}$ | load utilization index. |
| $\overleftarrow{Q}_{tfa}$ | load transfer from the front axle to the rear axle of the tractor (kN). |
| $h_{th}$ | height of the tow-hook (m). |
| $p$ | pitch of the tractor (m). |
| $\overleftarrow{Q}_{cd}$ | constant load due to the coupling device with load cells applied to the tractor rear part (kN). |
| $\overleftarrow{Q}_1, \overleftarrow{Q}_2, \overleftarrow{Q}_3, \overleftarrow{Q}_4$ | load transfer components from the front axle to the rear axle (kN). |
| $d_{TaRa}$ | horizontal component of the distance between the tow-hook and the rear axle (m). |
| $d_{\Delta h}$ | difference between the heights of the line of traction and the rotation axis of the rear wheels (m). |
| $\overleftarrow{Q}_d$ | component of the driver's weight burdening on the rear axle (kN). |
| $d_{sra}$ | distance of the rear axle from the vertical line of the centre of the seat (m). |

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
