# Peer review of "Assessment of the Performance of Agricultural Tires Using a Mobile Test Bench"

_agriculture, doi:10.3390/agriculture13010087_

Round 1

Reviewer 1 Report

This paper develops a mobile test bench to test the performance of agricultural tires. The research is very meaningful, the experimental data is detailed, and the experimental process is reasonable.. Please revise the following suggestions.

1. The unit of volume is not uniform, so it is recommended that the unit of pump displacement be L instead of cm3.

2. Is the unit of rotary speed r/min or min-1?

3. The symbol meaning in the legenda of p8 should be marked with units

4. P12 When testing the tire deflection, will the applied load cause axle deformation and affect the tire deflection value?

Reviewer 2 Report

The article is devoted to the study of the performance characteristics of agricultural tires. The chapters "Introduction" and "Materials and Methods" are described in sufficient detail. There is no doubt that the article has scientific and practical significance, nevertheless, the authors should specify in the "Introduction" chapter the purpose and objectives of the research presented in this article, reflect the novelty of the research.

The introduction contains references and a brief description of similar studies, but does not describe the disadvantages of research data, on the basis of which the authors decided to create a mobile test stand and evaluate the traction force.

In the section "Results and discussion", the authors do not compare the results obtained with the results of the studies presented in the introduction.

It is necessary to reflect numerical results in the conclusions and annotations: what accuracy of measurements is provided by the test stand, what range of tests, and so on.

Reviewer 3 Report

In order to evluate the performance of agricultural tires under different conditions, this manuscript developed a test bench and conducted a series of test to examine the accuracy and relability of the test bench, which could provide reference for the study on agricultural tires . Specific comments are as follows:

1. The introduction is too lengthy, it is suggested to simplify it by summarizing the status review.

2. The design and selection of structure and working parametes of the MTB could be more specific.

3. In 2.7 Line254, what factors could be examined with the MTB?

4. There is no any data in the conclusion.
